# EDITHOI: A FRAMEWORK FOR HOI IMAGE EDITING WITH SELF-GENERATED SKELETON GUIDANCE

## ABSTRACT

Recently, there were remarkable advances in image editing tasks. This could be categorized into text-guided global editing, local editing, text-guided local editing. To resolve the flawed human generation problem in prior image editing models, we propose a novel skeleton and text-guided local editing framework, EditHOI. Our goal is to edit an image by synthesizing an object-interactive human in the image. To do this, our framework consists of two stages: the first stage generates object-interactive skeleton using diffusion-based module, while the second stage outputs a Human and Object Interaction (HOI) image based on skeleton and text guidance. For effective evaluation on a object-interactive skeleton, we designed joint parameter and two evaluation metrics; object interaction top-$n$ accuracy and skeleton probability distance. The excellent performance of our framework is demonstrated through experiments qualitatively and quantitatively. Lastly, we show its applicability such as user controllable editing, generating pseudo SMPL ground truth and scalability to human-to-human interaction. The corresponding code is available at `https://anonymous.4open.science/r/HOI_editing_image-43F1/`

## 1 INTRODUCTION

Imagine kicking a ball on a playground. You could kick the ball with your left or right foot. You might kick it gently like a pass, or you could kick it hard as if you aim to score a goal. If you edit an image of a soccer ball to become the image you imagine, can your imagination be the same as everyone else's? It is definitely not, since it is a highly ill-posed problem. There are plenty of possibilities how human could interact with objects. To realize your imagination among various scenarios, we define Human and Object Interaction(HOI) image editing task and proposed a novel framework.

Text-guided global editing [1, 2, 3] basically edit images using an input prompt and an image. Models designed for this task alter the style of an image, apply colorization or generate objects based on textual prompts. However, as illustrated in the first row of Figure 1, we observed absence of human and incomplete multi-person generation, which can be critical in HOI image editing. Next, local editing models [4, 5, 6] are designed to fill in masked areas in consideration of the contexts of their surroundings. We attempted HOI editing using local editing models with a bounding box which represents the expected location of a person. However, as depicted in the second row of Figure 1, absence of human is observed, which can lead to a critical issue in HOI editing. Text-guided local editing models [7, 8, 9, 10] fill in a mask of an image, using both surrounding contexts and a text prompt. We tried HOI editing with text-guided local editing models using a text prompt and a person's bounding box. Unlike former methods, its performance looks relatively fine. However, four problems are still observed in third row at Figure 1. First, absence of human that a human is not generated. Second, incomplete human generation that improper human is generated. Third, absence of interaction that a human not interacting with object is generated. Last, incomplete multi-human generation that more than two people are generated improperly. As stated above, we could find the flawed human generation problem in prior works, which our framework overcomes with additional skeleton guidance.

We propose a novel skeleton and text-guided local editing framework which generates a skeleton interacting with an object and then uses this skeleton to inpaint local area in the image. Our framework

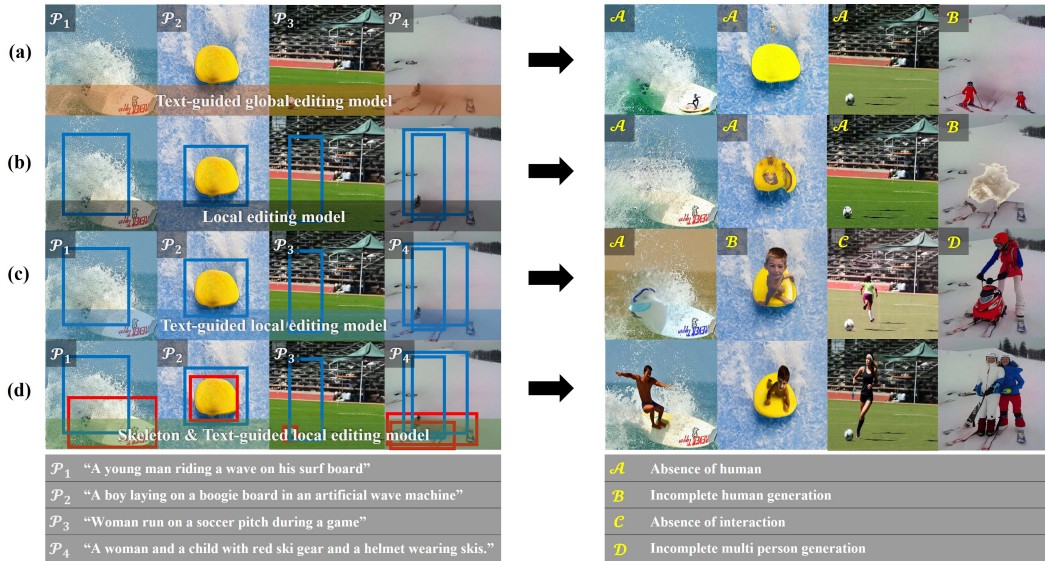

Figure 1: **Comparison of previous models with ours** : Images edited by (a) text-guided global editing, (b) local editing, (c) text-guided local editing, and (d) ours(skeleton and text-guided local editing). In the left Figure, the input condition and prompts($P_1$, $P_2$, $P_3$, $P_4$) are visualized. As shown in the right Figure, four problems, such as (A) Absence of human, (B) Incomplete human generation, (C) Absence of Interaction, (D) Incomplete multi-person generation, are observed in existing models. Our results exhibit more object-interactive human images.

consists of two stages. At first stage, a skeleton is generated with a diffusion-based object interaction skeleton generation module. Unlike fields of HOI classification [11, 12] or scene-interactive human motion generation [13, 14], in which only restricted indoor scenes are applicable, our approach can utilize various in-the-wild images to generate a skeleton interacting with object. At the second stage, the skeleton-guided editing model synthesizes object-interactive human on the masked image, using skeleton of first stage and text prompt.

Our approach solves these four existing problems in Figure 1. In the left side of Figure 1, the last row exhibits edited images using our framework. The first image shows that our framework have solved the absence of human problem that a human is not generated. This is because our framework directly generate a skeleton guidance. The second image shows that the incomplete human generation problem does not occur using our framework. The third image demonstrates the problem of absence and incomplete interaction is solved using our framework. The fourth image shows that our framework solved incomplete multi-person generation problem.

We also discover the potentials of our skeleton and text-guided local editing framework: EditHOI. In existing methods, there are two options for users when the results of image editing are unsatisfying. First, adjusting random seed iteratively until they obtain a satisfying output. Second, trying various prompts until the desired image is generated. Compared to previous works, our framework is more controllable, allowing users to edit the self-generated skeleton as the way they want. This makes it possible to generate a user-desired output using EditHOI. Additionally, more aligned pseudo SMPL [15] ground truth can be generated, since it can be optimized by SMPLify [16] using our self-generated skeleton. Moreover, a skeleton generated by our framework can be applied to human-to-human interaction.

We summarize our contributions below.

- We are the first to address the task of HOI image editing, synthesizing object-interacting realistic human on an image containing objects.

- We propose a novel skeleton and text-guided local editing framework, EditHOI. Our framework solved four problems in HOI image editing, demonstrated to outperform existing image editing models through experiments quantitatively and qualitatively.

- We suggest a diffusion-based object interaction module which generates object-interactive skeletons by itself. Additionally, we introduce new metrics and joint parameters for effec-

tive evaluation on a object-interactive skeleton. The effectiveness of the module and joint parameters is shown in ablation study.

- Our self-generated skeleton could be applied in various ways. Since our framework consists of two stages, users can choose to use the self-generated skeleton without modifications or manually edit it in order to get the desired output. More aligned pseudo SMPL [15] ground truth optimized by SMPLify [16] can be constructed with our self-generated skeleton. In addition, there appears to be the potential of scalability to human-to-human interaction

## 2  RELATED WORKS

**Text-guided global editing :** Text-guided global editing is a task which modifies the whole input image using text prompts. Specially, text prompts are provided in the form of a single noun or a combination of multiple words [17, 18, 19, 20], a sentence [21, 22, 23], and an instruction[1, 2, 3]. Style-CLIP [18] applied Contrastive Language-Image Pre-training(CLIP) models to StyleGAN, which enables intuitive text-guided image editing without additional manual controls. On the other hand, VQGAN-CLIP [23] is the first work to introduce a unified framework for both semantic image generation and image editing based on text prompts. Furthermore, Text2LIVE [17] extended its work of image editing to video. Rather than directly generating the output image, its key idea is to generate an edit layer which can be synthesized over the original image. However, as Text2LIVE [17] is designed to edit existing objects, it shows certain limitations in generating new objects. Instruct-Pix2Pix [1] edits an input image using user-provided instructions which inform the model of what to do. In the process of preparing a large-sized dataset of image editing examples, it integrated a language model (GPT-3) and a text-to-image model (Stable Diffusion) [24]. In addition, MagicBrush [2] introduces the first large-scale and manually annotated dataset for instruction-guided real image editing. It fine-tuned aforementioned Instruct-Pix2Pix [1] on MagicBrush [2] and demonstrated their new model achieves better results through human evaluation. A framework to utilize human feedback in instruction-guided image editing was introduced by Hive [3]. It obtained human feedbacks from annotators and fine-tuned diffusion models based on collected human preferences. Despite the significant advances in text-guided global editing, previous works are limited to replacing objects, changing the color of an image or modifying the background. Creating new objects remains either impossible or poorly processed in text-guided global editing.

**Local editing** : Local editing is a task which aims to edit the input image locally, filling in the masked or removed space in the image. Numerous works have achieved high-quality image synthesis quality with no guidance except for the surrounding contexts in the image, also known as inpainting [4, 5, 6, 25, 26, 24, 27, 28, 29, 30]. In the early stages of applying deep learning to inpainting task, [29] proposed a generative model that utilizes surrounding context around masks. With free-form mask and guidance, [28] presents a generative image inpainting network based on gated convolutions. For handling large-scale masks, CoModGAN [6] proposes co-modulated generative adversarial networks, a new method to reduce the gap between image conditional and unconditional GANs [31]. While existing inpainting models lack a large effective receptive field, LAMA [4] suggests an architecture called large mask inpainting (LaMa). MAT [5] integrates the merits of transformers and convolutions for large hole inpainting and high-resolution image generation. Similar to text-guided global editing, it is nearly impossible to create new object by local editing models, as they rely only on the surrounding contexts to fill in the missing region.

**Text-guided local editing** : Text-guided local editing is a relatively recently introduced task in computer graphics, filling in missing regions of an input image in consideration of both the surrounding context and additional textual descriptions [32, 33, 34, 8, 7, 10, 9]. Paint by Word [32] is the first method to investigate the problem of local zero-shot semantic image editing by pairing CLIP [35] with StyleGAN2 [36] and BigGAN [37]. GLIDE [7] utilizes diffusion models in two-stage approach for text-guided local editing : the first text-conditional diffusion model generates a row-resolution version of image, while the second stage processes upsampling using both the low-resolution version and the text prompt. SD-Inpainting [10] is a inpainting version of Stable Diffusion, while advanced Stable Diffusion XL [38] performs inpainting in SDXL-Inpainting [9] as well. BDM [34] was suggested by Avrahami et. al., performing local editing based on a textual description and an ROI [39] mask. It combined a Contrastive Language-Image Pre-training (CLIP) [35] model and a Denoising Diffusion Probablistic Model (DDPM) [40] to control the edit using a user-provided text prompt and generate generic natural images. Developing the prior work, Avrahami et. al proposed BLDM [8] ,

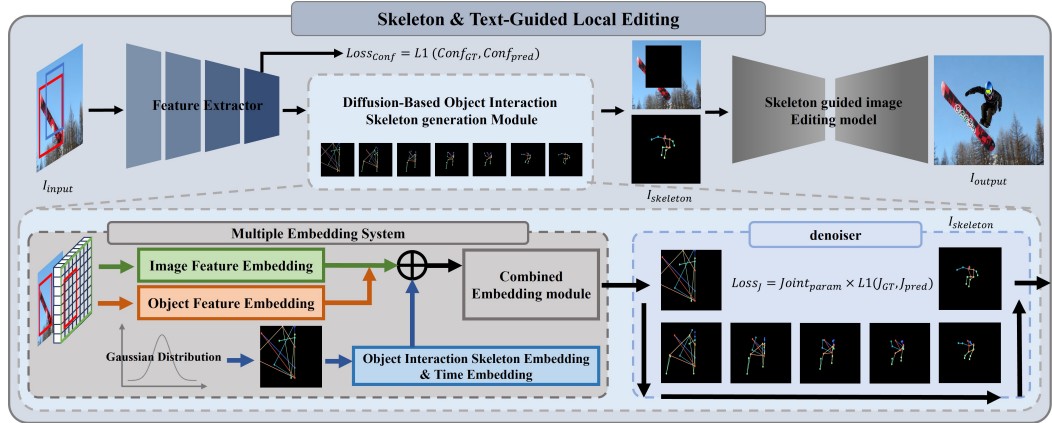

Figure 2: **Overview of proposed framework** : Our framework uses ResNet [41] backbone to extract features of an input image. In multiple embedding system, feature embeddings of an image and an object are embedded through an embedding network, using bounding boxes of a person and an object. Merging these embeddings with a noisy skeleton sampled from Gaussian distributions, combined embedding module feed the merged output to the denoiser network. Finally, a denoiser network reconstructs a skeleton interacting with an object.

an architecture with a text-to-image Latent Diffusion Model (LDM) [24] that works much faster than BDM [34]. Even though text-guided local editing is advancing rapidly, existing works still take a lot of time, making them hard to apply to real-time applications. More seriously, they fail to interpret text prompts describing not-familiar objects or scenarios, which causes failure in human and object interaction addressed in our paper.

Unlike previous works mentioned above, we suggest a framework employing additional skeleton guidance interacting with objects in the input image. With our method, higher-quality images can be generated than with prior works.

## 3  PROPOSED METHOD

At this section, we explain the overall process of our skeleton and text-guided local editing framework. Given an image, bounding boxes of a person and an object, a skeleton guidance is generated after passing them through a feature extractor and diffusion-based object interaction skeleton estimation module. Using the self-generated skeleton with the aforementioned inputs, our framework outputs the edited image which contain a human interacting with objects

### 3.1  SKELETON & TEXT-GUIDED LOCAL EDITING ARCHITECTURE

At this section, we propose a two-stage architecture for HOI image editing. The overall framework is visualized in Figure 2. and it consists of feature extractor, diffusion-based interaction skeleton estimation module and skeleton guided image editing model.

**Feature extractor :** We use a bounding box to define boundaries of an object interacting with a person. Without specifying locations of a person and an object, the network would struggle with where to locate the person and the object. Therefore, shown at the left side of Figure 2., we locate a person's bounding box $\mathcal{B}_{person} \in \mathbb{R}^{1 \times 4}$ as a hard decision. After that, the input image is fed to the backbone network.

Extracting a feature map from a backbone network, we obtain confidence score $Conf_{pred}$ which indicates whether each skeleton is visible or not. The ground truth confidence score of joints is 0 if the joint is invisible and 1 if the joint is visible. The predicted confidence is supervised by the ground truth confidence as followings:

$$Loss_{conf} = |p - \hat{p}| \tag{1}$$

where, $p$ and $\hat{p}$ indicate the ground truth confidence and estimated confidence respectively.

**Diffusion-based object interaction skeleton estimation module :** The feature map $F_{backbone}$ is fed through the multiple embedding system (MES). Shown in the left bottom of Figure 2., an image embedding $E_{image} \in \mathbb{R}^{17 \times 32}$ and an object embedding $E_{object} \in \mathbb{R}^{17 \times 32}$ are obtained from the feature map. We apply a region of interest (ROI [39]) pooling to the feature map with the object bounding box $\mathcal{B}_{object} \in \mathbb{R}^{1 \times 4}$ to obtain $F_{object} \in \mathbb{R}^{2048 \times N \times N}$ by the object feature embedding network. Next, using Gaussian distribution, we generate noisy joint $\mathcal{J}_{noised} \in \mathbb{R}^{17 \times 2}$. Adding those embeddings and $\mathcal{J}_{noised}$ together, we obtain $E_{skel} \in \mathbb{R}^{17 \times 32}$. And we add time embedding for timestep $T$ to get $E_{time} \in \mathbb{R}^{17 \times 32}$. All these embeddings together, using combined embedding module $\mathcal{C}$, we obtain object interaction noised skeleton $\mathcal{J}_{embedding} \in \mathbb{R}^{17 \times 2}$.

$$\mathcal{J}_{embdding} = \mathcal{C}(E_{image} \oplus E_{object} \oplus E_{skel} \oplus E_{time}) \tag{2}$$

A denoiser network gradually denoise $\mathcal{J}_{embedding}$ with timestep $T$ to obtain a object interactive skeleton $J_{pred} \in \mathbb{R}^{17 \times 2}$. An initial joint-wise L1 loss would guide predicted joints close to the ground truth joints. Moreover, we do not use this initial joint-wise L1 loss in naive manner. We consider how close the joint in ground truth is to the object bounding box, using our joint distance parameter $Joint_{param}$ as below:

$$Loss_{joint}^{\text{init}} = |J - \hat{J}| \tag{3}$$

$$Joint_{param} = \text{softmax}\left( \frac{1}{\text{dist}(J, \text{center}(\mathcal{B}_{object}))} \right) \tag{4}$$

$$Loss_{joint} = \lambda \times Joint_{param} \times Loss_{joint}^{\text{init}} \tag{5}$$

where $J$ and $\hat{J}$ indicates the ground truth joint and the predicted joints respectively. And center$(\cdot)$ is a function which computes the center location of a bounding box. We use Euclidean distance to measure distance between the center of the object bounding box and a joint location. We update initial joint-wise loss using $Joint_{param}$ as a weight. $\lambda$ is a scale factor we set $\lambda = 10^{-4}$ in experiments.

After that, we choose D3DP [42] as a diffusion structure which reconstruct noisy joint at timestep $T$.

**Skeleton-guided Image Editing Model :** In this stage, we edit an input image $\mathcal{I} \in \mathbb{R}^{H \times W \times 3}$ using $\mathcal{I}_{skel} \in \mathbb{R}^{H \times W \times 3}$ and $\mathcal{B}_{person}$ as additional conditions. We modify the predicted skeleton from the previous network to MSCOCO [43] format. We note that skeleton guidance to $\mathcal{I}_{skel}$.

Unlike previous image editing model, we use generated $\mathcal{I}_{skel}$ as a condition. Using our $\mathcal{I}_{skel}$, we could solve the aforementioned four problems of previous editing models. In addition, users could modify our predicted joints manually which is impossible in previous models. Moreover, changing a object bounding box to a person bounding box generates skeletons interacting each other. Using the predicted skeleton to generate pseudo SMPL [15] ground truth demonstrates its applicability. We employ ControlNet-Inpainting [44] as the skeleton guided imaged editing model.

## 4 EXPERIMENTS

At this section, we compare our method with existing methods in qualitative and quantitative ways, demonstrating the effectiveness of our framework. We conduct ablation study to show the effectiveness of our newly developed $Joint_{param}$ for object interaction. Moreover, we experiment various methods to obtain a human skeleton: (1) using feature embeddings of an image and an object and MLP, (2) using these embeddings and GNN, (3) using these embeddings, Gaussian noise and diffusion algorithm. We demonstrate that (3) works most effectively compared to the others. Implementation details are on our supplementary materials.

### 4.1 DATASETS

V-COCO [45] is well-known dataset in HOI field. Different from datasets such as HICO [46] and Bongard-HOI [47], it contains ground truths of segmentation, skeletons and a person's bounding

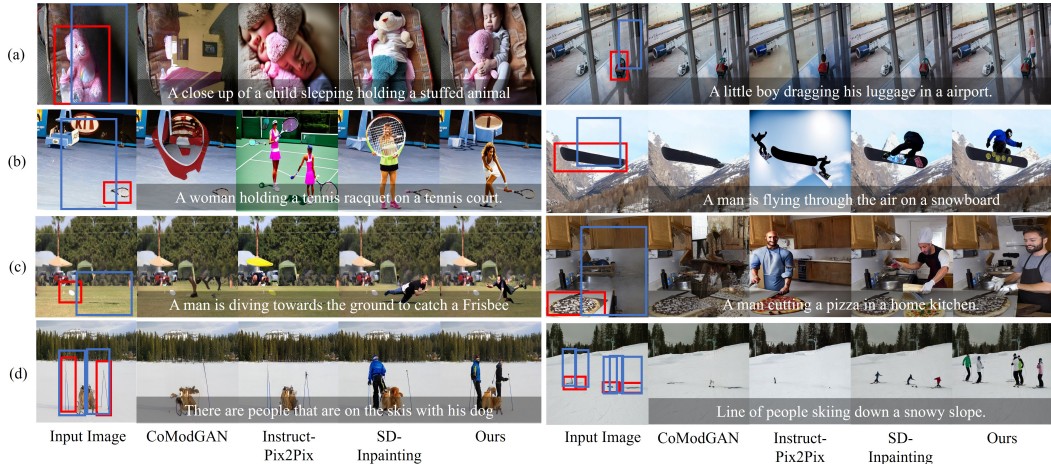

Figure 3: This figure shows comparison of three editing frameworks to ours. CoModGAN [6], Instruct-Pix2Pix [1], Stable-Diffusion Inpainitng (SD-Inpainting) [10] were used for comparison. The results of CoModGAN[6] and Instruct-Pix2Pix [1] failed to generate a human in most cases. In edited images using SD-Inpainting [10], aforementioned four problems occurred in order: (a) absence of human, (b) incomplete human generation, (c) absence of interaction, (d) incomplete multi-person generation

box. We made a masked area based on the segmentation ground truth of a person and filled the mask using LAMA [4]. To select images containing HOIs, we collected images of which Intersection over Union (IoU) are greater than zero. We used V-COCO [45] protocol for training and testing.

## 4.2 EVALUATION METRIC

To quantitatively compare our framework with existing methods, we use Frèchet Inception distance (FID [48]), Kernel Inception distance (KID [49]) and CLIP score (CS [50]) as evaluation metrics.

**Frèchet Inception Distance (FID [48])** : FID [48] aims to compare the distributions of generated images to images from a real dataset. Assuming two datasets follow Gaussian distribution $\mathcal{N}(\mu, \Sigma), \mathcal{N}\hat{\mu}, \hat{\Sigma})$, FID [48] is defined as:

$$\text{FID}(\mathcal{N}(\mu, \Sigma), N(\hat{\mu}, \hat{\Sigma})) = ||\mu - \hat{\mu}||_2^2 + \text{Tr}(\Sigma + \hat{\Sigma} - 2(\Sigma\hat{\Sigma})^{1/2}) \quad (6)$$

**Kernel Inception Distance (KID [49])** : KID [49] measures the squared maximum mean discrepancy (MMD) between the feature of inception network of the real and generated images using a polynomial kernel. Since it is a non-parametric test, it does not need the strict Gaussian assumption.

**CLIP score (CS [50])** : CS [50] measures the extent to which the generated images are aligned with the text conditions. In precise manner, it is a metric that represents the extent to which a text condition matches an images without relying on human annotations. Let $I$ be an input image, $C$ be a corresponding text condition, and $E_I, E_C$ be embeddings within the image and text condition, respectively. Then, the CLIP score [50] is defined as follows :

$$\text{CLIPScore}(C, I) = \max(100 \times \cos(E_C, E_I), 0) \quad (7)$$

where the CLIP score [50] is between [0, 100].

FID [48] and KID [49] are indicators of how generated image is realistic. And CS [50] measures how well synthesized image is well aligned with a prompt describing interactions. Moreover, to evaluate the generated skeleton interacting with object, we define two evaluation metrics.

**Object interaction top-$n$ accuracy** : This metric represents the extent to which the interacting joints in the generated image are similar to interacting joints in the real world. Specifically, it is 1 when the predicted joint is inside the object bounding box where its index is same as the $n^{th}$ closest joint in the ground truth skeleton to the object bounding box and 0 otherwise.

**Skeleton Probability Distance (SPD)** : SPD measures the extent to which the joints interacting with an object are similar to the real world data. The IoU of object bounding box and the bounding box covering joints is calculated. This IoU is computed for the bounding box covering ground truth joints and estimated joints. The size of bounding box is a manually defined. The joint-wise calculated IoUs are normalized by softmax. And a distance between normalized joint-wise IoUs of ground truth and estimated joints is computed with Jensen-Shannon distance [51]. The SPD of bounding boxes of ground truth joints $\mathcal{B} = \{B_i\}$ and bounding boxes of predicted joints $\hat{\mathcal{B}} = \{\hat{B}_i\}$ is defined as:

$$\text{SPD}(\mathcal{B}, \hat{\mathcal{B}}; \mathcal{B}_{object}) = \text{dist}(\text{softmax}(\text{IoU}(\mathcal{B}_{object}, \mathcal{B})), \text{softmax}(\text{IoU}(\mathcal{B}_{object}, \hat{\mathcal{B}}))) \qquad (8)$$

## 4.3 Quantitative Results

Table. 1 shows quantitative results on various editing models. Text-guided local editing and skeleton and text-guided local editing models show the best performance on average among text-guided global editing, local editing, text-guided local editing, skeleton and text-guided local editing models. Our framework skeleton and text-guided local editing model outperform others. Our framework uses the same diffusion backbone of SD-inpainting and improved 4.14 in FID [48], 0.0035 in KID [49] and 1.22 in CS [50] than vanilla SD-inpainting [10]. Moreover, SD-inpainting [10] using our framework outperforms SDXL-

Table 1: **Quantitative results comparing our framework to the previous image editing model** : our framework outperform others on the metrics indicating image quality FID [48], KID [49] and metric measuring prompt alignment to image CS [50].

| Comparision Editing Model | | | |
|---|---|---|---|
| Evaluation Metric | FID [48] (↓) | KID [49] (↓) | CS [50] (↑) |
| Text-Guided Global Editing Model | | | |
| Instruct-Pix2Pix [1] | 45.37 | 0.0200 | 28.44 |
| MagicBrush [2] | 60.01 | 0.0381 | 28.89 |
| HIVE [3] | 56.38 | 0.0346 | 27.70 |
| Local Editing Model | | | |
| LAMA [4] | 59.30 | 0.0342 | 27.08 |
| MAT [5] | 77.55 | 0.0479 | 21.87 |
| CoModGAN [6] | 52.30 | 0.0282 | 26.18 |
| Text-Guided Local editing model | | | |
| Glide [7] | 63.14 | 0.0344 | 25.70 |
| BLDM [8] | 25.52 | 0.0090 | 29.06 |
| SDXL-Inpainting [9] | 25.01 | 0.0082 | 29.63 |
| SD-Inpainting [10] | 28.16 | 0.0087 | 29.24 |
| Skeleton & Text-guided Local editing model | | | |
| **SD-Inpainting [44] + Ours** | **24.02** | **0.0052** | **30.46** |

inpainting [9] which is an enhanced model of SD-inpainting [10]. This demonstrates the significance of our framework in HOI image generation.

## 4.4 Qualitative Result

Figure 3. is the qualitative comparison between our models to Instruct-Pix2Pix [1] which is a text-guided global editing model, CoModGAN [6] which is a local editing model, stable diffusion inpainting (SD-Inpainting) [10] which is text-guided local editing model. In most cases, CoModGAN [6] and instruct pix2pix [1] do not generate human properly. So we concentrate on comparing with SD-inpainting [10]. (a) shows the absence of humans in generated images. In the case of SD-inpainting [10], a child and a little boy are contained in the prompts but no human is generated. (b) shows the generation of incomplete or awkward human and object interaction. In the case of SD-inpainting [10], the generated tennis racket and a person overlapped which could not happen in the real world. On the right side of (b) only black objects are generated except for ours. (c) shows the results of a human and an object not well interacting. In the case of SD-inpainting [10], there is no interaction between a person and a frisbee but ours interact well. In addition on the right side of (c), despite the prompt including the phrase "cutting a pizza" our image is the only one that generates proper interaction. Last, (d) shows the examples of other models that do not generate multi-person properly. However, our model uses additional explicit skeleton guidance so that generates natural images with multi-person. Additional comparisons between SDXL and ours are on supplementary materials.

## 4.5 Ablation Study

Table. 2 is the results from our model in the presence and absence of our proposed $Joint_{parm}$. We experiment with ResNet [41] 50, 101, 152 as the backbone network with using a object bounding

Table 2: **Quantitative results on the absence and presence of proposed** $Joint_{param}$ : Left side of the arrow shows the result without using $Joint_{param}$ while right side shows the results using it. Overall results enhanced using our proposed parameter.

| ResNet [41] Backbone Comparision Using Our Hyper parameter | | | | |
|---|---|---|---|---|
| Backbone | Object interaction | | | Skeleton |
| evaluation | Top 1 (↑) | Top 3 (↑) | Top 5 (↑) | distance (↓) |
| ResNet [41] 50 | 58.1 → 58.9 % | 64.6 → 65.1 % | 67.3 → 68.2 % | 0.135838 → 0.132561 |
| ResNet [41] 101 | 58.9 → 60.8 % | 65.3 → 67.2 % | 67.8 → 68.6 % | 0.131394 → 0.130857 |
| ResNet [41] 152 | 56.8 → 58.6 % | 62.8 → 64.8 % | 65.4 → 67.3 % | 0.133987 → 0.131039 |

**(a) Anomaly Pose Problem**

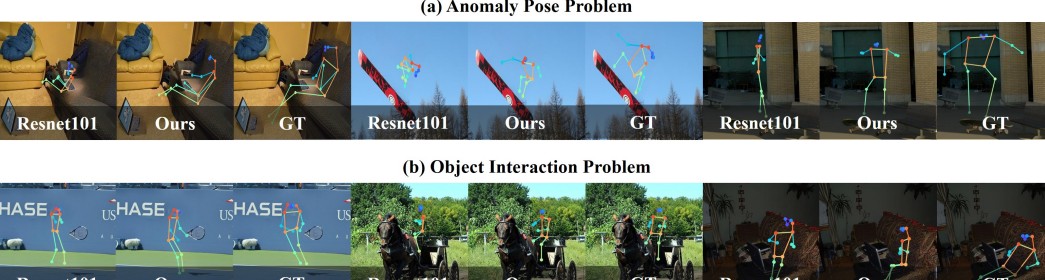

**(b) Object Interaction Problem**

Figure 4: This figure shows the results of absence and the presence of our diffusion-based object interaction skeleton estimation module generating a human skeleton. (a) show the problem that generating a skeleton where joints are squeezed which is an anomaly pose. (b) shows the problem of generating invalid interaction with an object.

box. Every results in object interaction top 1, 3, 5 have increased for all backbone. Moreover, skeleton probability has been increased through all backbone networks.Especially ResNet [41] 101 scores highest in perspective of object interaction. After these, we experiment predicting joints with MLP, GNN and diffusion with image and object embeddings using the object bounding box. Shown in Table. 3, the results using MLP and GNN are worse than the naive models in Table. 2 As a result, the model using diffusion achieve the best results among them. We not only quantitatively compare them but also visualize the results at Figure 4. We visually compare our model with the model using ResNet [41] 101, since the model using ResNet [41] 101 is the best among others except for our model.

Two major problems are shown in Figure 4. on the models without using our framework using the ResNet [41] 101 as a backbone network. First is the anomaly pose problem. Second is the object interaction problem. Our framework using diffusion module, gradually de-noise on the noisy skeleton so that obtain plausible skeleton. However, us-

Table 3: **Quantitative results applying** $E_{object}$ **on various methods** : using diffusion based method shows the best quantitative results than others.

| Comparison Object Embedding Module | | | | |
|---|---|---|---|---|
| Method | Object interaction | | | Skeleton |
| evaluation | Top 1 (↑) | Top 3 (↑) | Top 5 (↑) | distance (↓) |
| MLP | 60.8% | 67.2% | 68.6% | 0.130857 |
| GNN | 58.6% | 64.7% | 67.2% | 0.131156 |
| **Diffusion** | **62.6%** | **68.2%** | **70.6%** | **0.126317** |

ing only the ResNet [41] 101 to predict joint shows non-interactive skeletons generation shown at the Figure 4. bottom, since its architecture is not complex enough to consider object embeddings well. Theses shows that our diffusion-based object to object interaction human pose estimation module is effective.

## 4.6 USER STUDY

We survey using the image generated by the text-guided global editing model, local editing model, text-guided local editing model and our proposed framework on image generation quality, prompt relevance, image with the best editing and image well interacting with the object. 83.2% of people agree that the image quality generated with our framework is better than others. And 85.7% of people think that our framework has the highest prompt relevance. 86.7% of people think our framework shows the best image editing quality. 83.6% of people agree that our framework is the best model showing plausible interaction with an object. Considering that these four criteria have significant meaning in image editing, we conclude that 84.8%

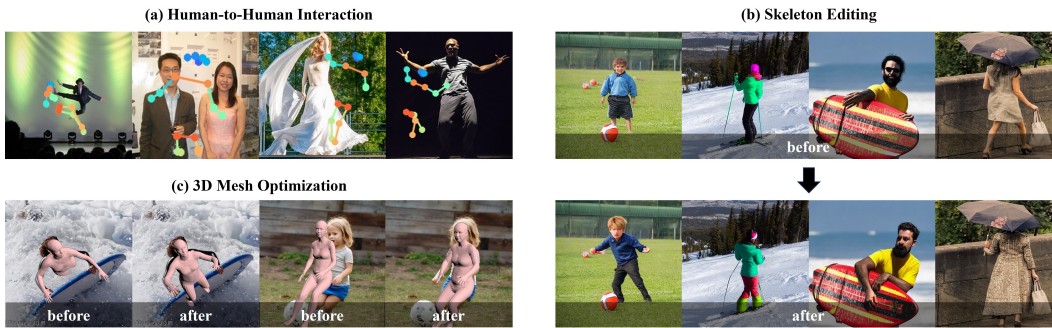

Figure 5: This figure shows three application cases using our framework. (a) demonstrates its extension to human-to-human interaction, (b) shows that manual editing of the predicted skeleton could enhance the image quality better. And (c) shows the results of 3D mesh optimization using SMPLify [16].

## 5  APPLICATIONS

In this section, we show that our framework could be extended or applied to various tasks. Shown in Figure 5. (a), we confirmed the potential for expansion from object-to-human interaction to human-to-human interaction. We experiment with its possibility by simply changing the object bounding box to the person bounding box. We were able to get a skeleton who dances, step on people and surprise. Using this skeleton guidance, we generated images with our framework but failed to synthesize plausible images. This is because using a person bounding box for masking eliminates most part of the given images. So, there would be not enough information to infer. Additionally, under trained diffusion model might be the reason. We left these problems to our future work to solve.

Next, we could manually edit the skeleton shown in Figure 5. (b). Most editing models heavily rely on prompts so we have to modify prompts elaborately or might change random seeds until get what we want. However, using our framework we obtain an estimated skeleton from the network and users could manually modify these skeletons to what they want. So, a more elaborate modification is possible. This solves the heavy reliance on the prompt that existing editing models have.

Finally, obtaining 3D human mesh is possible shown in Figure 5. (c). Owing to the recent development of image generative models, powerful data augmentation tools were used in face-related datasets [52]. These development has shown promise in a variety of task such as hand and human pose. However, most editing or generative models only rely on prompts to generate images. This is the critical problem of an existing model. Because if the result is unsatisfying then users should accept or reject the output and there is no other option. Or they might compromise to use them even if there is a misalignment with the prompt. However using our framework to optimize 3D human mesh with SMPLify [16], we would obtain a much elaborate and precise pseudo 3D human mesh dataset. This technique is a well-known method in the 3D human mesh estimation field. These extensive applications are our strength.

Our framework could be developed in various fields and is more practical than existing editing models. We believe that the development of this technology will have a huge impact on the field of computer vision in the future.

## 6  CONCLUSIONS

In this study, we define a HOI image editing task and propose a novel framework for HOI image editing, EditHOI. Our framework solves the four critical problems in existing editing models by generating skeleton guidance to edit an image by itself. We demonstrated that our framework outperforms than the others in quantitative and qualitative ways. In addition, we show the potentials of our framework in new applications. Although our framework is still limited to object interaction, it can be applied to the fields of human-to-human and human pose estimation in future developments.

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
