## A    SUPPLEMENTARY MATERIALS

### A.1    DETAILED OF USER STUDY

Figure 6 shows a comparison of text-guided local editing, text-guided global editing, local editing, and ours. It visualizes the results for image quaility prompt relevance, image with the best editing, and image well interacting with object. We could see that our framework overwhelmingly outperformed the other framewors. Moreover, Fig 7 shows a questionnaire via goole-form. (a), (b), (c) and (d) show a rando mix of the four frameworks, and a total of 15 visualizations are shown.

### A.2    ADDITIONAL OBJECT INTERACTION SKELETON QUALITATIVE RESULTS

8 shows additional visualization results which are not on the main paper. In case of the results of anomaly pose problem for example, images lying on a bed or jumping on a bench or using a laptop, unknown skeleton is generated but not in the results using our framework. We could obtain better object interacting skeleton with object feature embedding than without using it so that more natural skeletons are generated which better describe the situation.

### A.3    ADDITIONAL OBJECT INTERACTION IMAGE COMPARISION SDXL QUALITATIVE RESULTS

We only demonstrate qualitative results by SD-inpainting [10] as a comparison in our main paper. We more qualitative results with SDXL [9]. Similarly, using SDXL [9] there were four problems, (a) absence of human, (b) incomplete human generation, (c) absence of interaction, (d) incomplete multi person generation. In the first row of figure 9, we have an absence of human problem where the multi person disappears or the chef preparing the food is not properly visible. In the second row, we have an incomplete human generation where the woman in bed is not properly created or the person skiing is not properly created. In the third row, we have an absence of interaction where the wooden sppon is not properly interacted with or the tennis racket is not properly interacting with. In the last row, we have an incomplete multi person generation problem where the multi person is not properly created. However, you could see taht our method sovles all of these problems.

### A.4    IMPLEMENTATION DETAIL

We use Pytorch [53] in training and evaluating our framework. Moreover, we use ImageNet pre-trained ResNet backbone from torchvision [53]. And we use Adam optimizer [54] in training with batch size 64. Initially, we set learning rate to $10^{-4}$ and reduce by $\frac{1}{10}$ at epoch 70 and 120. A single Nvidia RTX-3090 is used to train and inference our framework. We use D3DP [42] for diffusion-based object interaction skeleton estimation module. A simple MLP network is used in combined

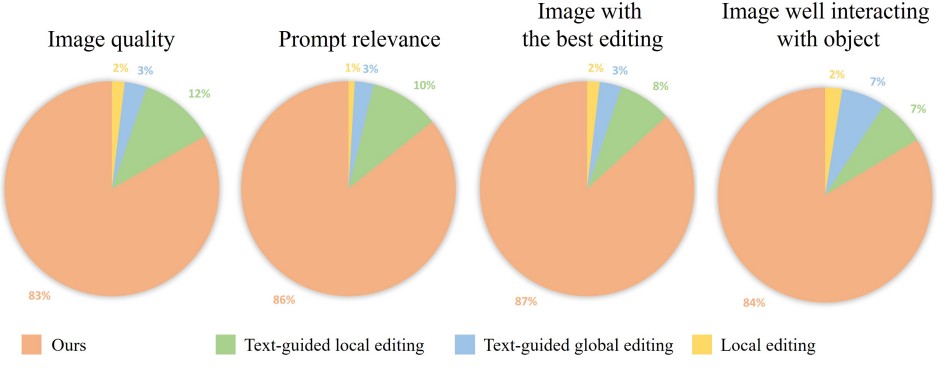

Figure 6: A user study result showing that our framework outperform others in all four categories: image quality, prompt relevance, image with the best editing and image well interacting with object.

1. Please select one of the images (a), (b), (c), or (d) that you feel has the highest score. [The goal is to edit the input image to fit the red box. I would appreciate it if you could choose from this point of view.]

"A group of people getting onto a bus carrying surfboards"

| | Input Image | (a) | (b) | (c) | (d) |

| | (a) | (b) | (c) | (d) |
|---|---|---|---|---|
| Image quality | ☐ | ☐ | ☐ | ☐ |
| Prompt relevance | ☐ | ☐ | ☐ | ☐ |
| Image with the best editing | ☐ | ☐ | ☐ | ☐ |
| Image well interacing with object | ☐ | ☐ | ☐ | ☐ |

Figure 7: Examples of our survey format. Surveyee can choose which is better for each of the four categories.

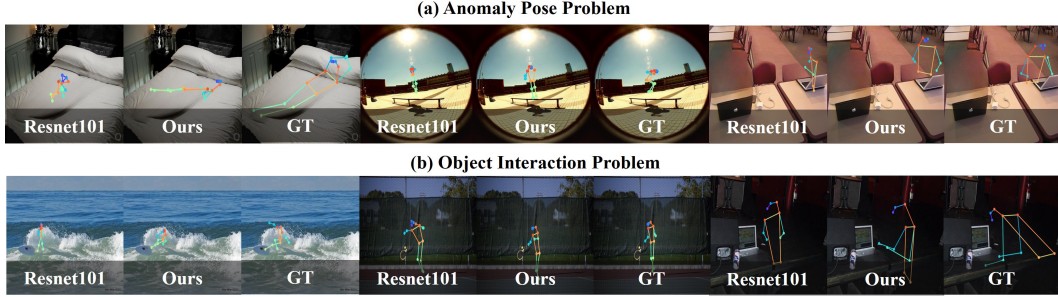

**(a) Anomaly Pose Problem**

| Resnet101 | Ours | GT | Resnet101 | Ours | GT | Resnet101 | Ours | GT |

**(b) Object Interaction Problem**

| Resnet101 | Ours | GT | Resnet101 | Ours | GT | Resnet101 | Ours | GT |

Figure 8: This figure shows the results of absence and the presence of our diffusion-based object interaction skeleton estimation module generating a human skeleton. (a) show the problem that generating a skeleton where joints are squeezed which is an anomaly pose. (b) shows the problem of generating invalid interaction with an object.

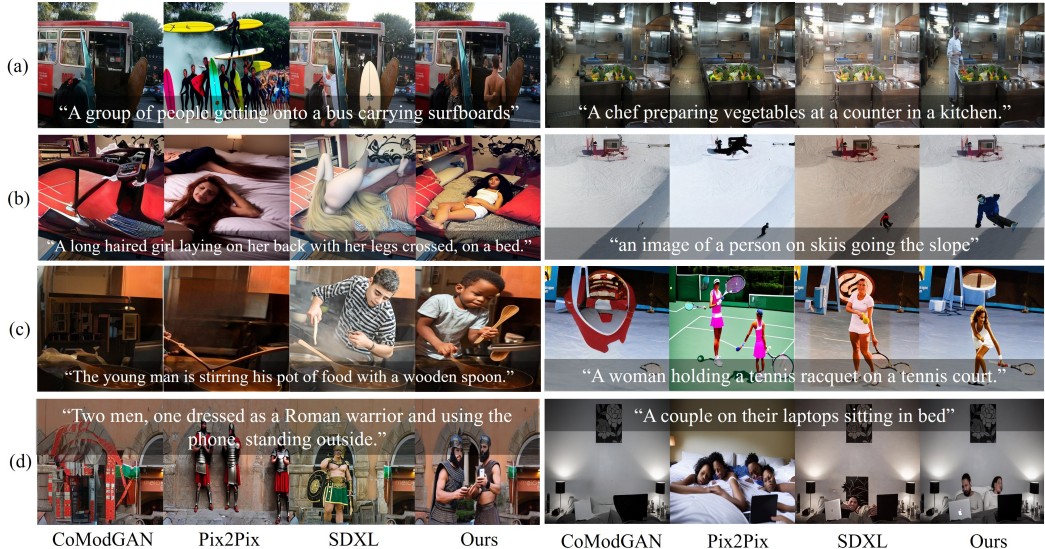

Figure 9: This figure demonstrate the comparison between representative editing framework to ours. CoModGAN [6], Instruct-Pix2Pix [1], Stable-Diffusion XL inpainitng (SDXL) [9] were used for comparison. Compare with SDXL [9], (a) shows the absence of human that needs to be generated. (b) shows the incompleteness of human generation. (c) shows the absence of object interaction. (d) shows the results of incomplete generation of multi-person.

embedding module. In addition, various time embeddings were used. In inference stage, we set timestep to 2000 and employ denoiser. Stable-diffusion controlnet inpainitng [44] is used for skeleton guided image editing model.