# OpenReview forum: "EditHOI: A framework for HOI image editing with self-generated skeleton guidance"
_ICLR.cc/2024/Conference — ICLR 2024 Conference Withdrawn Submission_

### Official Review · Reviewer_tHfv · 2023-10-29

**Soundness:** 1 poor
**Presentation:** 2 fair
**Contribution:** 1 poor
**Rating:** 1
**Confidence:** 5

**Summary:**

The goal of this paper is to edit an image by synthesizing an object-interactive human in the image. To do this, their framework consists of two stages: the first stage generates object-interactive skeleton using diffusion-based module, while the second stage outputs a Human and Object Interaction (HOI) image based on skeleton and text guidance. However, their only contribution is to train a network that predicts skeleton based on the diffusion model.

**Strengths:**

- Good writing. This paper is well written and easy to follow.

**Weaknesses:**

- Limited innovation and contribution. The only contribution of this paper is to train a network that predicts skeleton based on the diffusion model, which is not novel.
- Insufficient experiments. Lack of comparison with HumanSD [1].

  [1] Ju, Xuan, et al. ‘HumanSD: A Native Skeleton-Guided Diffusion Model for Human Image Generation’. Proceedings of the IEEE/CVF International Conference on Computer Vision (ICCV), 2023, pp. 15988–15998.

- Incorrect statements and unfair comparisons. For example, there are ablation experiments on diffusion model vs. ResNet101 that do not provide comparisons of parameters and inference times, which is clearly unfair; also, both quantitative and qualitative experiments in comparisons with other methods illustrate precisely the power of ControlNet rather than this method.

Therefore, the paper is not at all up to the standard of ICLR and I recommend the authors to choose another conference.

**Questions:**

As HumanSD found out, ControlNet still has all four types of problems, and you only generate conditional inputs for ControlNet-Inpainting, so why do you dare to claim that you have solved all four types of problems? Besides, it has nothing to do with your contribution.

---

### Official Review · Reviewer_rWaG · 2023-10-31

**Soundness:** 3 good
**Presentation:** 1 poor
**Contribution:** 3 good
**Rating:** 6
**Confidence:** 3

**Summary:**

The authors propose the first method for the generation of Human-Object-Interaction (HOI) images. Specifically, they introduce EditHOI composed of two stages. In the first stage, they design a diffusion-based module to generate an object-interactive skeleton based on the pre-defined bounding box. Next, the authors will have the second stage to generate the HOI images based on skeleton and text prompts. Beyond the methods, the paper provides two novel evaluation metrics for better evaluations. Both qualitative and quantitative results showcase its capability to successfully generate HOI images with reasonable human actions, surpassing the existing state-of-the-arts.

**Strengths:**

The strengths of the proposed paper can be summarized as:
1. The authors propose a novel skeleton and text-guided local editing framework that can generate Human-Object-Interaction images based on a single input text prompt.
2. Both qualitative and quantitative evaluations demonstrate the capability and superiority of the proposed method.

**Weaknesses:**

The weaknesses of the proposed paper can be summarized as:
1. The qualitative results presented in Figure 1 and 3 are not obvious and good enough, although it indeed surpass the existing methods.
2. There is no visualization for the generated skeleton and analysis on this part. More visualization and explanation would be beneficial.
3. What the results will look like if we remove the bounding-box conditions?

**Questions:**

N/A

---

### Official Review · Reviewer_jZPv · 2023-10-31

**Soundness:** 2 fair
**Presentation:** 2 fair
**Contribution:** 1 poor
**Rating:** 5
**Confidence:** 3

**Summary:**

The paper introduces "EditHOI," a novel framework for Human and Object Interaction (HOI) image editing, addressing the complex and ill-posed nature of image editing tasks. EditHOI stands out in its ability to synthesize object-interactive humans into images, a task that has been problematic for existing image editing models due to issues like the flawed generation of humans, absence of interaction, and incomplete multi-human generation.

**Strengths:**

EditHOI introduces a novel approach to the task of HOI image editing, an area that has not been extensively explored. The framework's two-stage process, which involves generating an object-interactive skeleton and using it for image inpainting, is a creative solution to the problem of synthesizing object-interacting humans in images. This represents an inventive combination of diffusion-based models and text guidance for image editing.

The framework is methodologically sound, employing a diffusion-based module for skeleton generation that can handle diverse and unstructured in-the-wild images. The use of text prompts in the second stage to guide the image synthesis process demonstrates a thoughtful integration of multimodal inputs (skeletal structure and textual description) for image editing.

**Weaknesses:**

Technical contribution: the technical contribution of this work is limited. The main part involves generating an object-interactive skeleton with diffusion models and using it for image inpainting.

Scalability: It aims to handle a very specific editing task, i.e., Human and Object Interaction(HOI) image editing. The narrow scope of this topic significantly limits its generalization across different tasks.

Failure Analysis: While the paper discusses the framework's success in overcoming certain problems in HOI image editing, a detailed analysis of failure cases would be beneficial.

**Questions:**

see above.

---

### Official Review · Reviewer_BnY7 · 2023-11-07

**Soundness:** 2 fair
**Presentation:** 3 good
**Contribution:** 2 fair
**Rating:** 3
**Confidence:** 3

**Summary:**

This paper aims to edit an image by synthesizing an object-interaction human in the image, which first generates skeleton using the diffusion-based module, and second outputs a human and object interaction image based on skeleton and text guidance

**Strengths:**

- This paper proposes a technical pipeline dedicated to addressing the issue of poor performance of HOI in the community of text-to-image.

**Weaknesses:**

The proposed solution appears technically rigorous, yet lacks novelty.
- The resolution to the four issues raised by the authors: absence of human, incompleteness of human, absence and incompleteness of interaction, and incomplete multi-person generation problem, seems to be significantly attributed to ControlNet.
- The skeleton generation network based on the diffusion model lacks innovative elements. Utilizing various embeddings to guide diffusion models is a common technique, such as the employment of text embedding, time embedding, and image embedding to guide the text-to-image model.

**Questions:**

- Generating HOI is a challenging task. Is there a more fundamental solution to enhance the Text-to-image model’s ability to generate interaction, rather than simply combining existing algorithmic techniques, such as ControlNet and SD-Inpainting?
- Control generation based on skeletons is a common solution for images with high dependence on structure and layout, typically involving initial skeleton generation followed by image generation. Given that skeletons can be correctly generated based on interactive objects, why can’t the subject be directly generated from interactive objects? Is it feasible to not use skeletons as an intermediate bridge? It would be beneficial if this could be elucidated through experimentation.